# A Narrative Review on Gut Microbiome Disturbances and Microbial Preparations in Myalgic Encephalomyelitis/Chronic Fatigue Syndrome: Implications for Long COVID

**DOI:** 10.3390/nu16111545

**Published:** 2024-05-21

**Authors:** Joanna Michalina Jurek, Jesus Castro-Marrero

**Affiliations:** 1Unit of Research in Myalgic Encephalomyelitis/Chronic Fatigue Syndrome and Long COVID, Rheumatology Research Division, Vall d’Hebron Research Institute, Universitat Autònoma de Barcelona, 08035 Barcelona, Spain; jesus.castro@vhir.org; 2Grup de Recerca GEMMAIR (AGAUR)-Medicina Aplicada (URV), Departament de Medicina i Cirurgia, Institut d’Investigació Sanitària Pere Virgili, Universitat Rovira i Virgili, 43005 Tarragona, Spain

**Keywords:** probiotics, gut microbiota, gut–brain axis, chronic fatigue syndrome, myalgic encephalomyelitis, long COVID, prebiotics, dietary supplementation, antioxidants, inflammation, psychobiotics

## Abstract

Myalgic encephalomyelitis, also known as chronic fatigue syndrome (ME/CFS), and long COVID are complex, multisystemic and long-term disabling conditions characterized by debilitating post-exertional malaise and other core symptoms related to immune dysregulation resultant from post-viral infection, including mitochondrial dysfunction, chronic neuroinflammation and gut dysbiosis. The reported associations between altered microbiota composition and cardinal symptoms of ME/CFS and long COVID suggest that the use of microbial preparations, such as probiotics, by restoring the homeostasis of the brain–immune–gut axis, may help in the management of symptoms in both conditions. Therefore, this review aims to investigate the implications of alerted gut microbiome and assess the evidence supporting use of microbial-based preparations, including probiotics, synbiotics, postbiotics alone and/or in combination with other nutraceuticals in the management of fatigue, inflammation and neuropsychiatric and gastrointestinal symptoms among patients with ME/CFS and long COVID.

## 1. Introduction

Myalgic encephalomyelitis, also referred to as chronic fatigue syndrome (ME/CFS), is a complex and long-term debilitating condition affecting ~67 million people worldwide, characterized by persistent fatigue not improved by rest or sleep, and post-exertional malaise (PEM) resultant from minor physical or mental activity, consequently leading to declines in cognitive impairment, including problems with concentration, thinking and memory [1,2]. In addition, individuals with ME/CFS may also experience a constellation of other non-specific symptoms of pain, headaches, disrupted sleep, myalgia and arthralgia, orthostatic intolerance, autonomic dysfunction and gastrointestinal (GI) problems that greatly interfere with their ability to function and significantly compromise individual productivity and quality of life [2,3,4]. Although there is no specific diagnostic test available for ME/CFS, the diagnosis of this condition is made through medical history, physical examination and laboratory tests, along with clinical assessment, including use of agreed diagnostic criteria [5], which requires a presence of four key symptoms for a period of at least 6 months, including debilitating fatigability, post-exertional malaise, unrefreshing sleep and cognitive difficulties [6], possibly accompanied by other manifestations, of which the most common are orthostatic intolerance and autonomic dysfunction (e.g., dizziness, palpitations, fainting, nausea when standing or sitting upright); pain (e.g., myalgia, headaches, abdominal pain or joint pain); neuromuscular symptoms (e.g., twitching and myoclonic jerks); flu-like symptoms (e.g., sore throat, tender glands, nausea, chills or muscle aches); and hypersensitivities to temperature, certain foods and sensory exposures of light, noise, touch and smell [7].

Complex and non-specific symptomatology of ME/CFS may lead to many misdiagnoses, with other serious illnesses, including fibromyalgia, primary sleep disorders (sleep apnea), nutritional disorders (iron deficiency, obesity), musculoskeletal conditions, anxiety/depression and GI diseases (celiac disease, irritable bowel syndrome (IBS) and inflammatory bowel disease (IBD)) [8]. In addition, recent studies conducted shortly after the onset of the COVID-19 pandemic suggested that symptoms of ME/CFS are similar to those reported by approximately 87% of patients recovering from acute infection with SARS-CoV-2 [9,10], which significantly interfered with their ability to function and execute daily tasks—a condition referred to as long COVID [10]. Despite a lack of evidence of major injuries in organs caused by the infection, these individuals have been shown to experience 25 out of 29 known ME/CFS symptoms at least once, also including those considered as the key features used by the diagnostic criteria for ME/CFS, including reduced daily activity, post-exertional malaise (PEM) and disabling fatigue being the dominant ones [11]. Interestingly, long COVID patients have also reported other dysfunctions that previously were associated with neurologic pain, neurocognitive and psychiatric symptoms (however, with no cases of rash) and olfactory and gustatory dysfunction [11] (Figure 1).

Although having the exact etiology of ME/CFS, long COVID still needs to be determined. Evidence accumulated to date indicates that post-viral syndromes most likely triggered by infectious agents, such as Epstein Barr virus and herpesviruses in ME/CFS [12] and SARS-CoV-2 in long COVID [13], respectively, also share biological abnormalities, with examples of cognitive deficits, including impaired attention and information processing speed, dysregulation of the hypothalamic–pituitary axis (HPA) and an abnormal immune cytokine profile [14], characterized by high autoantibody titers against neural and autonomic targets [15,16], including neurotransmitter receptors against nuclear and membrane structures, such as cardiolipin and phospholipids, neurotransmitter receptors (e.g., muscarinic M1 acetylcholine receptor (AChR) and ß1- and ß2-adrenergic receptors (AdR) and M2/3 [17]); increased levels of pro-inflammatory mediators (such as IL-1α, IL-1β, IL-4, IL-5, IL-6, IL-12, TNF-α, IL-10, IL-13, IL-16, INF-γ and IL-17, IL-17A) but reduced IL-17F [18,19,20], further linked with severity of mentioned symptoms [21]. In particularly, metabolic abnormalities along with mitochondrial dysfunction and impaired redox balance [22] characterized by increased oxidative toxicity and lowered anti-oxidant defenses were associated with the development of symptoms of pain and hypersensitivity [23] and severity of neuropsychiatric symptoms in both ME/CFS [18] and long COVID [24]. Noteworthy, mitochondrial dysfunction, as an important contributor to both ME/CFS and long COVID conditions [25,26] through increasing production of free radicals, can promote chronic inflammation, which, along with aberrant immune responses reported in ME/CFS and COVID-19 [27], attributes to diminished natural killer cell function, T cell exhaustion/abnormalities, overactivation of mast cell along associated with possible viral reactivation, triggered by viral infection/infectious agent may increase risk of autoimmunity and hyperinflammation [28], including systemic inflammation and neuroinflammation [29], which are associated with muscle weakness [30], fatigue and pain [31] and other comorbidities, such as major depression [32], postural tachycardia syndrome [33] and GI conditions, like development allergic reactions and sensitivities to odors, chemicals and foods including alcohol, caffeine, sweeteners and food additives [34]. Some studies suggest that long COVID patients may acquire new food allergies and sensitivities (especially to wheat and gluten), with the incidence of GI symptoms ranging from 3% to 79% [35]. Similarly, GI symptoms are also considered a frequent comorbidity for up to 92% of ME/CFS patients who are co-diagnosed with irritable bowel syndrome (IBS) [2], while GI complaints, including abdominal pain and bloating, along with extra-intestinal symptoms of fatigue, headache and cognitive impairments, could be developed due to chronic gut inflammation and compromised intestinal epithelium barrier [36], followed by a significantly increased proportion of mucosal associated invariant T (MAIT) cells [37], which are implicated in the recognition of microbial antigens in the MHC Class I-related molecule [38] and the response to cytokine-induced stimulation during microbial infection [39].

Furthermore, the proposed hypothesis indicates that persistent infection or unviable pathogen residues can stimulate chronic inflammation, as both enteric viruses and bacterial infections can affect the microbiome [40]. Alteration in the composition of the gut microbiota in both long COVID and ME/CFS patients along with reported GI symptoms of nausea, diarrhea and abdominal pain [41] may suggest a potential role of gut dysbiosis in the development of both conditions. Research to date have shown that gut dysfunction, characterized by altered intestinal microbiota composition, gut barrier dysfunction and gut inflammation, can be a potential factor in ME/CFS development [3]; whereas, for long COVID patients with persisting symptoms of inflammation and poor cognitive performance for a few months after acute infection, alterations in the gut microbiota along with reduced microbial diversity are linked to disease severity and reported by patients with long COVID symptoms, including lung, all supporting a potential contribution of disturbed gut microbiome in long COVID sequelae [42].

Microbial preparations, not limited to the use of live probiotic strains, but also their by-products alone or in combination with multivitamins and/or other bioactives, such as flavonoids and indigestible dietary fibers (prebiotics), due to their immunomodulatory properties, may offer potential benefits for individuals affected by long COVID and ME/CFS, particularly because both conditions lack effective treatment strategies [43,44]. In addition, there is growing evidence to demonstrate that the intake of live (probiotics) or inactivated microorganisms is confirmed beneficial for human health (if given in appropriate amounts [43]) by promoting gut health and immune homeostasis and may help to reduce inflammation and cognitive dysfunction in ME/CFS [45]. Similarly, combined co-supplementation of both probiotics and prebiotics, in the form of synbiotics such as the blend of the *Lactobacillus* strain with inulin have been shown to improve health outcomes of individuals with long COVID, leading to reduced cough, fatigue and gut symptoms [46].

Given the possible role of microbiome dysbiosis in shared manifestations reported in long COVID and ME/CFS, along with the beneficial effects of therapeutic interventions with microbial preparations on restoring gut homeostasis, these might serve as a novel approach for the management of shared ME/CFS and long COVID symptoms, including chronic inflammation and cognitive and bowel dysfunctions. The main aim of this review is to investigate the implications of alerted gut microbiome and assess the evidence supporting the use of microbial-based preparations, including probiotics, synbiotics, postbiotics alone and/or in combination with other nutraceuticals in the management of fatigue, inflammation and psychiatric and gastrointestinal symptoms in patients with ME/CFS and long COVID.

## 2. Role of the Alerted Gut Microbiome in ME/CFS and Long COVID Pathogenesis

The gut microbiota is a tightly regulated microbial community of more than 10^14^ microorganisms [47], including a variety of archaea, viruses, fungi and protists, which, in humans, are dominated by the phyla Firmicutes, Bacteroidetes, Proteobacteria and Actinobacteria [48]. The microbiota works in partnership with the host immune system to fulfill essential roles for optimal health and well-being by directly supporting immunity and metabolism, and through bioactive compounds, including short-chain fatty acids (SCFA) (e.g., acetate, propionate and butyrate); amino acids (e.g., γ-aminobutyric acid (GABA); and vitamins, acting on the gut–brain axis, which actively modulate digestive and cognitive functions [49]. The gut microbiota is also an essential component of the host’s defense mechanisms against infections, additionally contributing to the maintenance of intestinal homeostasis and gut mucosal barrier integrity [50]. Nevertheless, exposure to various factors, including diet, medications/antibiotics and certain lifestyle habits (e.g., smoking and alcohol) can influence gut microbiota composition, leading to the colonization of opportunistic pathogens and intestinal barrier dysfunction. Imbalance of the gut microbiota community, also referred to as gut dysbiosis, is linked to increased intestinal permeability and tight junction dysfunction, allowing the entry of commensal microbes, microbial-derived products and other luminal components [51], which contribute to aberrant immune responses and increase the risk of infections and chronic diseases, including obesity, diabetes type II, IBD and cardiovascular disease (CVD) [52].

Reported alterations in the gut composition from patients with ME/CFS and long COVID warrant attention, as they are associated with both conditions’ symptoms of fatigue, post-exertional malaise (PEM), pain, sleep disturbances, mental and GI disturbances and immunological abnormalities [11]. Although these observations may suggest the role of gut dysbiosis in symptom development [53], a specific microbial signature has not yet been found [2]. Investigations of the microbiome dysbiosis have been intensively studied in ME/CFS, with results indicating a potential link between changes in gut composition, including deficient butyrate-producing capacity and bacterial network disturbances, increased immune activation [54,55,56] and symptoms of fatigue symptoms [2,57]. The decreased availability of butyrate, along with other key mediators of the gut–brain axis produced during microbial fermentation, including serotonin and γ-aminobutyric acid (GABA) via the influence of intestinal permeability, may have considerable impact on the neuro-immunoendocrine system [58]. In addition, studies from physically fit populations have indicated a strong correlation between increased gut microbial diversity, high fecal butyrate levels and extent of physical fitness, characterized by higher VO_2_ peaks and, consequently, reduced LPS levels and lower inflammation [59], thereby suggesting that the decreased abundance of anti-inflammatory bacteria, such as Firmicutes and Actinobacteria [54] along with butyrate-producer bacteria, may result in PEM followed by inactivity and increased LPS-induced inflammation in ME/CFS. In addition, ME/CFS patients, compared with healthy controls, seem to also have higher levels of pro-inflammatory Gram-negative bacteria species, such as Alistipes, Bacteroidetes and Enterobacteriaceae [54,56,60], which may worsen the disease course, leading to an increased severity of fatigue, pain and confusion [54]. In addition, ME/CFS has been associated with the reduction of certain bacteria taxa of the Firmicutes phylum, including *Faecalibacterium*, *Roseburia* and *Clostridium* genera, of which, a decrease in the butyrate-producer Faecalibacterium was considered a potential biomarker with diagnostic value in ME/CFS [56]. Noteworthy, associations between a reduced abundance of *Faecalibacterium* and increased fatigue perception were found in IBD, cancer and autoimmune patients, such as MS and diabetes type 1 [2], thereby suggesting that the presence of systemic inflammation along with gut microbiota dysbiosis [61] may affect the bidirectional connection with other vital organs [62] and lead to the development of neuropsychiatric and GI symptoms, also reported in ME/CFS. Previous studies have shown that reductions in the *Firmicutes* Gram-positive bacteria, which include beneficial lactic acid bacteria such as *Actinobacteria* (e.g., Collinsella and *Bifidobacterium* spp. *Bifidobacteria*), at the expense of Gram-negative bacteria, such as *Bacteroidetes*, and *Proteobacteria*, can alter immune response, promote pro-inflammatory cytokine production and increase gut barrier permeability, further leading to chronic inflammation [63], which in the case of ME/CFS, has been implicated in the development of cognitive dysfunctions, sleep problems, mood changes and various physical symptoms like muscle aches, flu-like symptoms and weight changes [45]. Therefore, a reported presence of gut inflammation in ME/CFS may contribute to neuroimmune dysfunction followed by the gradual activation of innate responses in the brain via the vagus nerve and a reduction in energy-consuming activities in ME/CFS [2]. For instance, a reduced abundance of *Actinobacteria* reported in ME/CFS [64] was previously associated with GI conditions, including IBD and IBS, and neuropsychiatric diseases, such as anxiety/depression. Interestingly, ME/CFS patients who reported IBS and high anxiety/depression symptoms had a significantly lower alpha diversity of gut microbiome compared with healthy controls and IBS-only cohorts, suggesting that comorbid IBS and anxiety/depression may be linked to higher *Proteobacteria*, *Prevotellaceae*, *Bacteroides* and lower *Lachnospiraceae* abundances relative to controls [65]. This complex crosstalk between the HPA axis, immune factors and the autonomic nervous system highlight the role of a dysregulated gut–brain axis in the establishment of low-grade gut mucosal barrier inflammation, followed by increased visceral hypersensitivity and increased intestinal permeability (also known as leaky gut) in ME/CFS [2].

Previous evidence obtained during the COVID-19 pandemic has revealed that gut dysbiosis is associated with the severity of acute SARS-CoV-2 infection and long-lasting multisystem complications after disease recovery [66], with a currently growing number of studies reporting changes in the intestinal composition and diversity in long COVID patients who recovered from acute infection. These individuals have been shown to have significant changes in gut composition, characterized by the reduction in commensal species remaining low for up to 30 days, including *Eubacterium* rectale, *Faecalibacterium* prausnitzii and bifidobacterial species (e.g., *Lactobacillus*, *Bifidobacterium*, *Bacteroides* and *Faecalibacterium*), whereas the richness of the microbial community seems to not recover even after 6 months of convalescence [67,68,69]. Noteworthy, the reduced abundance of *Faecalibacterium prausnitzii* and the decrease in anti-inflammatory species, *Alistipes onderdonkii* and *Faecalibacterium prausnitzii*, reflected by increased levels of pro-inflammatory cytokine, including IL-2, IL-7, IL-10 and TNF-α, is strongly associated with disease severity and increased severity of symptoms, including respiratory, neuropsychiatric, gastrointestinal and fatigue issues [70,71]. In addition, an observational study conducted in patients who developed long COVID symptoms 6 months after initial diagnosis (76% of an initial sample of 106 patients) had significantly higher levels of *Ruminococcus gnavus* and *Bacteroides vulgatus* and lower levels of *Faecalibacterium prausnitzii*, which were correlated with persistent respiratory symptoms and neuropsychiatric complaints and fatigue, including *Clostridium innocuum* and *Actinomyces naeslundii*. Interestingly, butyrate-producing bacteria, including *Bifidobacterium pseudocatenulatum* and *Faecalibacterium prausnitzii*, showed the largest inverse correlations with long COVID symptoms at 6 months [72]. Interestingly, a prospective follow-up study, conducted on symptomatic patients recovered from COVID-19, reported a presence of certain gut microbiota dysbiosis, including significantly reduced bacterial diversities and lower relative abundance of SCFAs producers, such as *Eubacterium hallii*, *Subdoligranulum*, *Ruminococcus*, *Dorea*, *Coprococcus* and *Eubacterium ventriosum*, a year after discharge [73]. Interestingly, individuals experiencing long-term symptoms of physical function impairment, psychiatric disorders (mainly anxiety or depression), lung function reduction and radiographic abnormalities persisted for 12 months in individuals with altered gut microbiota was significantly correlated with clinical indices of the recovery stage, thereby suggesting that gut microbiota may play an important role in long COVID [66]. 

Taken together, emerging evidence suggests that gut dysbiosis is associated with the regulation of brain activity and cognitive function in ME/CFS and long COVID via the microbiota–gut–immune–brain axis. However, it is unknown to date how the gut microbiome and neuroglia cells might interact with each other and how these interactions might trigger the onset of neurocognitive symptoms (also known as “brain fog”) and post-exertional fatigue in ME/CFS and long COVID. We hypothesize that one of the possible triggers could be dysregulation in the production, transport and functioning of neurotransmitters and their specific receptors (reduced excitability of GABA-ergic and glutamatergic activity within the primary M1 motor cortex) [74,75,76].

## 3. Use of Probiotic in the Management of ME/CFS and Long COVID Symptoms

Probiotics and prebiotics are the two components in our diet that can affect the microbiome [66]. Probiotics, as live microorganisms that are administered in adequate amounts [77] utilizing the non-digestible fibers of prebiotics, can bring health benefits to the host [66], including improved digestion and immunity, and, by modulating immune signaling pathways and enhancing gut microbiota function within the gut–lung and gut–brain axes [78], reduce common symptoms of infection [79] and increase resistance to the infectious agent. In addition, probiotic preparations can help with the management of neurodegenerative and psychiatric comorbidities related to low mood, sleep problems, depression and anxiety-like symptoms [80,81] and those associated with chronic GI conditions, including IBS [82].

Probiotics have also been evaluated in the context of ME/CFS, as previous studies have shown that they can lead to a significant improvement in anxiety/depression symptoms and promoting overall well-being. In addition, immunomodulatory and anti-inflammatory properties of certain probiotic strains have been shown to reduce inflammation and lower oxidative stress in ME/CFS patients [83,84], consequently leading to improved cognitive function and reduced fatigue severity. In addition, metabolites of microbial origin, by acting on the gut–brain axis, can influence brain function [85], with favorable effects on mood and sleep and reduced depression, anger and fatigue [86]. Similar benefits were also observed in hospitalized COVID patients, who, after receiving probiotic supplementation, especially with *Lactobacillus* and *Bifidobacterium* strains [87], had reduced overall symptoms, including lowered inflammatory reactions, decreased hospitalization duration and recovery time, which may suggest that probiotics may have the potential to reduce mortality resultant from this condition [88].

Examples of studies conducted on both ME/CFS and COVID-19 cohorts investigating the effect of interventions based on probiotic strains in the context of the most common symptoms reported, such as inflammation and fatigue and neuropsychiatric and GI symptoms, are presented in Table 1.

### 3.1. Effects of Probiotics on Fatigue and Inflammation in ME/CFS and Long COVID

Post-infectious fatigue is a common complication linked with chronic/abnormal inflammation that can significantly decrease physical and mental efficiency, cause depression/anxiety, perturb sleeping patterns and overall decrease quality of life. Evidence to date has demonstrated the beneficial role of probiotics and their metabolites in modulating immunity and reducing inflammation and damage arising from oxidative stress. This can be especially important in reducing severe fatigue and exertion intolerance linked with ongoing chronic inflammation resulting from abnormal immunity and mitochondrial dysfunction [53,98]. Consequently, the use of probiotics in the treatment of ME/CFS and long COVID has been intensively studied, owing to their potential to reduce inflammation and restore immune homeostasis, while promoting immunity and recovery [66] might be beneficial for reducing their shared symptoms of post-infectious fatigue and leading to improved overall health.

Interventions conducted in patients with ME/CFS focused predominantly on the use of Bifidobacteria and Lactobacillus strains, which consistently demonstrated good efficiency in reducing inflammation and oxidative stress, and were accompanied by a reduced perception of fatigue. For example, the use of probiotic treatments with *Lactobacillus casei* strain Shirota over 8 weeks significantly reduced inflammatory markers in ME/CFS patients [89]; whereas, an oral intake of *Bifidobacterium infantis* 35624 for 8 weeks decreased concentrations of pro-inflammatory markers, including CRP and IL-6, compared with the baseline in 70% of ME/CFS patients, thereby suggesting that *B. infantis* 35,624 benefits are not limited to mucosal responses and they can exert an effect systemically [99]. In addition, use of multi-strain probiotic formulations, which previously have been shown to counteract antibiotic-resistant pathogens, strengthen mucosal barrier and regulate immune responses, including preparations *Lactobacillus rhamnosus* combined with *Lactobacillus casei* (Ramnoselle) and *Lactobacillus casei* combined with *Bifidobacterium lactis* (Cytogenex) that were effective after 8 weeks in ameliorating fatigue (Chadler’s score), which has been attributed to reduced inflammation, which was reported as an almost 30% reduction in CRP levels compared with baseline, and enhanced immunity, characterized as a significant increase in IgM (three times over the baseline values) and a reduced CD4/CD8 ratio. It must be noted that the high variability between patients warrants further studies to justify the immunomodulatory potential of a probiotic regime [84]. Nevertheless, the supplementation of a multi-strain probiotic formulation containing *Lactobacillus paracasei* ssp. paracasei F19, *Lactobacillus acidophilus* NCFB-1748 and *Bifidobacterium lactis* Bb12 demonstrated a limited efficacy in reducing fatigue and disability scores among ME/CFS patients, as a 4-week-long treatment followed by 4 weeks of follow-up improved patient scoring on the SF-12 health survey in the area of neurocognitive functioning, there was no significant improvements in fatigue and physical activity scores assessed on the visual analogue scales (VAS) or major changes in the gut microbiota composition [90].

Probiotic supplementation has been shown to have potential in lowering systemic inflammation among patients with both acute infection and long COVID syndrome [100]. To date, interventions with probiotics conducted in COVID-19 patients, including strains *Lacticaseibacillus rhamnosus*, *Lactiplantacillus plantarum*, *Lactobacillus salivarius*, *Lactobacillus acidophilus*, *Pediococcus acidilactici*, *Bifidobacterium bifidum*, live *Bifidobacterium longum*, *Bifidobacterium longum* subsp, *Pediococcus acidilactici* and *Streptococcus thermophilus*, have been associated with positive health outcomes, including lowered systemic inflammation and reduction in serum CRP and improvement in respiratory symptoms, particularly cough and shortness of breath, however with no obvious effect on fever, headache or weakness [69]. For example, clinical intervention with a commercial probiotic (SLAB51), known as Sivomixx800^®^ (Ormendes, Switzerland), consisting of *Streptococcus thermophilus* DSM 32245^®^, *Bifidobacterium lactis* DSM 32246^®^, *Bifidobacterium lactis* DSM 32247^®^, *Lactobacillus* acidophilus DSM 32241^®^, *Lactobacillus helveticus* DSM 32242^®^, *Lactobacillus paracasei* DSM 32243^®^, *Lactobacillus plantarum* DSM 32244^®^ and *Lactobacillus brevis* DSM 27961^®^, has been shown to significantly lower the proportion of long COVID patients reporting fatigue on the FAS scale compared with individuals without the treatment after 3 weeks of treatment. This effect was further attributed to favorable changes in the key metabolites related to glucose metabolism among patients receiving oral bacteriotherapy, who had significantly increased concentrations of serum arginine, asparagine and lactate and lower levels of 3-hydroxyisobutirate than controls [91]. Similarly, treatment with a probiotic consortium of eight live and freeze-dried strains of *Streptococcus thermophilus* BT01, *B. breve* BB02, *B. animalis* subsp. *lactis* BL03, *B. animalis* subsp. *lactis* BI04, *L. acidophilus* BA05, *L. plantarum* BP06, *L. paracasei* BP07 and *L. helveticus* BD08 (VSL#3^®^) after 4 weeks significantly reduced fatigue scores on the Chalder fatigue scale (CFS) in long COVID patients compared with placebos, which were maintained for 4-week post-intervention [92]. Furthermore, combining probiotic strains into formulations of multi-ingredient preparations also including bioactive components efficient in reducing inflammation in respiratory infections [101] has been proposed as an efficient strategy in the management of long COVID symptoms. For example, a multi-enzyme formulation (ImmunoSEB) consisting of probiotic (ProbioSEB CSC3), containing a blend of *Bacillus coagulans* LBSC (DSM 17654), *Bacillus subtilis* PLSSC (ATCC SD 7280) and *Bacillus clausii* 088AE (MCC 0538) strains along with Peptizyme SP, including enteric coated serratiopeptidase, bromelain, amylase, lysozyme, peptidase, catalase, papain, glucoamylase and lactoferrin, significantly reduced total physical and mental fatigue scores in patients with COVID-19 compared with placebo. In addition, this supplement was well tolerated, with no adverse events reported. In addition, after the initial 14 supplementations, those patients with mild-to-moderate disease severity reported health benefits, demonstrated as reduced systemic inflammation determined as reduced CRP levels (by 77% in the treatment and by 56% for the placebo, respectively), and enhanced recovery was achieved on day 10 of the intervention compared with the placebo [86]. Similar benefits were also observed with the use of synbiotics combining probiotic strains of Lactobacillus plantarum, Lactobacillus rhamnosus, Lactobacillus bulgaricus, Lactococcus lactis and Lactobacillus paracasei with prebiotic inulin fiber, and a phytochemical-rich whole food blend of citrus sinensis fruit, chamomile (*Matricaria recutita* L. flower), curcuma longa, pomegranate (*Punica granatum* L.) and resveratrol extracted from Polygonum cuspidatum root, which, after 4 weeks, resulted in an almost two-fold reduction in the mean fatigue scores and a double improvement in the overall well-being scores on the subjective well-being score compared with the placebo [95].

Furthermore, a recent small study conducted on females co-diagnosed with fibromyalgia and ME/CFS were shown to be effective in reducing levels of perceived fatigue and the impact of fibromyalgia on these patients, compared with the baseline after symbiotic (Synbiotic, Gasteel Plus^®^ (Heel Spain, S.A.U.) supplementation, whose formulation was based on probiotics, e.g., *Bifidobacterium lactis* CBP-001010, *Lactobacillus rhamnosus* CNCM I-4036 and *Bifidobacterium longum* ES1, combined with prebiotics, fructooligosaccharides (200 mg) and micronutrients, including zinc (1.5 mg), selenium (8.25 g) and vitamin D (0.75 g) [96]. In addition, a pilot study investigating the effectiveness of multi-component preparation “OMNi-BiOTiC^®^STRESS Repair 9” (Institute AllergoSan, Graz, Austria), consisting of probiotics, e.g., *Lactobacillus casei* W56, *Lactobacillus acidophilus* W22, *Lactobacillus* paracasei W20, *Bifidobacterium lactis* W51, *Lactobacillus salivarius* W24, *Lactococcus lactis* W19, *Bifidobacterium lactis* W52, *Lactobacillus plantarum* W62 and *Bifidobacterium bifidum* W23; prebiotic fructooligosaccharides (FOS), such as inulin, enzymes (amylases) and selected micronutrients, including potassium chloride, vitamin B2 (riboflavin 5′-sodium phosphate), vitamin B6 (pyridoxine hydrochloride), manganese sulfate and vitamin B12 (cyanocobalamin), has been shown to be effective in improving fatigue, mood and quality of life in both the probiotic and placebo groups, however with greater improvements reported in the probiotic group after 6 months of treatment [97]. Although the results of these studies suggest that bacteriotherapy in the management of long COVID is promising, the low number of trials, often varying in the duration of treatment and tested formulation of the supplement (probiotic strains used and dose; presence of other bioactives, e.g., vitamins, minerals, phytonutrients) has not provided sufficient evidence to use microbial preparations in the clinical care of patients with post-viral syndromes.

The reported benefits of improved immunity, energy metabolism, respiratory functions and psychiatric well-being suggest that probiotic supplementation has the potential to improve the management of ME/CFS and long COVID. Although the results of conducted trials may suggest that supplementation with single- and multi-strain probiotics may have potential benefits for reducing fatigue and inflammation in ME/CFS and long COVID patients, only a few studies using the preparations of *Bifidobacteria* and *Lactobacillus* demonstrated their good efficiency in reducing inflammatory markers, and their effects on the fatigue scoring scales remains inconsistent, thereby indicating the need for further exploration into the interplay between probiotic bacteria and fatigue and inflammation in ME/CFS and long COVID.

### 3.2. Effects of Probiotics on Neuropsychiatric Symptoms (Psychobiotics)

Psychobiotics, as selected probiotic strains and their preparations, ingested in sufficient amounts may have positive impacts on mental health, including reduced symptoms of depression and anxiety and chronic stress, and improved sleep quality [102], attributed to the ability to modulate the gut–brain axis [103] and/or the production of neuroactive bioactives, such as γ-aminobutyric acid (GABA) and serotonin [104]. Although many probiotics have been proposed as potential psychotropic agents, including *Streptococcus thermophilus*, *Bifidobacterium animalis*, *Bifidobacterium bifidum*, *Bifidobacterium longum*, *Streptococcus thermophiles*, *Lactobacillus bulgaricus*, *Lactococcus lactis*, *Lactobacillus acidophilus*, *Lactobacillus plantarum*, *Lactobacillus reuteri*, *Lactobacillus paracasei*, *Lactobacillus helveticus*, *Lactobacillus rhamnosus*, *Bacillus coagulans*, *Clostridium butyricum* and others [104,105], depending on the acquired neuropsychiatric condition [105], their effectiveness may vary. In context of post-viral syndromes, such as ME/CFS and long COVID, specific strains of *Lactobacillus* and *Bifidobacterium* have gained interest, as, other than their immunomodulatory effects, they may also be relevant for reducing risk factors associated with these conditions, including low quality of life, disturbed sleep, mood changes, mental fatigue, anxiety/depression and chronic stress. Although none of the psychobiotic formulations were exclusively tested in ME/CFS and long COVID patients, there is growing evidence indicating that the use of probiotic strains might be beneficial for improving neurocognitive function and managing symptoms of stress and anxiety [104] frequently reported by these patients. For example, psychobiotic metabolites acting as neurotransmitters and neurochemicals can influence the brain–gut axis and modulate the stress response via the HPA axis. Stress, proposed as the factor linked with fatigue induction, can cause neuroendocrine deregulation resulting from the activation of the HPA axis and stimulation of glucocorticoid production, which has been characterized as increased cortisol levels and the secretion of pro-inflammatory cytokines. The chronic activation of the HPA axis can also significantly affect the gut microbiota composition, which has been linked to the presence of mental and physical illnesses. Therefore, the use of microbial preparations, such as psychobiotics aiming to restore gut homeostasis and normalize HPA axis hyperactivity, may help to relieve negative consequences of chronic stress and inflammation implicated in the pathogenesis of ME/CFS and long COVID [104,105].

To date, studies have shown that psychobiotics can help in reducing depressive symptoms and potentially can be used as supportive therapy in major depressive disorder (MDD). For example, *Clostridium butyricum* (CBM588) supplementation along with prescribed antidepressants significantly reduced (e.g., ≥ 50% reduction) a total HAMD-17 score in the Hamilton rating scale for depression and Beck depression inventory (BDI) scores in patients with treatment-resistant MDD compared with the placebo [106]. Similar benefits were reported in the management of generalized anxiety disorder (GAD) and anxiety-like symptoms with multi-strain preparations, including *L. heleveticus* Rosell-52 and *B. longum* Rosell-175 [107], *Bifidobacterium longum*, *Bifidobacterium bifidum*, *Bifidobacterium lactis* and *Lactobacillus acidophilus* [108], and a multi-strain consortium of *L. acidophilus* LA5 and *B. lactis* BB12 [109], which significantly reduced the intensity of anxiety symptoms and improved overall mental health outcomes. Additional benefits attributed to psychobiotic formulations consisting of *Lactobacillus fermentum* LF16, *L. rhamnosus* LR06, *L. plantarum* LP01, and *Bifidobacterium longum* BL04; *Lactobacillus acidophilus*, *Lactobacillus casei* and *Bifidobacterium bifidum*; *Lactobacillus helveticus* R0052, *Bifidobacterium* longum R0175 and *Lactobacillus plantarum* 299 have been shown to significantly improve mood by reducing depressive and angry feelings [110,111], and to reduce the response to stress on the HPA axis [112], observed as reduced salivary cortisol levels [93], all together leading to improved sleep. In particular, the supplementation of Lactobacillus paracasei HII01 in fatigued participants after 12 weeks has been shown to significantly reduce salivary cortisol levels compared with the baseline [93], suggesting potential benefits also for fatigue reduction among patients with post-viral syndromes.

While the use of psychobiotics to modify the gut microbiome has been shown to improve anxiety [107,108] and depression symptoms [106,110,113], and improve mental well-being upon chronic stress [109,114] with additional benefits for sleep quality and neurocognitive functioning [111], the direct application of probiotic strains with psychobiotic properties may have the potential to reduce psychiatric symptoms in ME/CFS and long COVID patients, owing to the reported gut dysbiosis along with aberrant immune responses and persistent systemic inflammation. For example, overlapping psychiatric symptoms between MDD, ME/CFS and long COVID suggest that treatments with certain psychobiotic strains among ME/CFS, such as *Lactobacillus casei* strain Shirota, through lowering inflammation, may help to reduce anxiety and improve cognitive function and overall functioning [89]. Similar benefits were also reported after the use of preparations consisting of Lactobacillus rhamnosus combined with *Lactobacillus casei* (Ramnoselle) and *Lactobacillus casei* combined with *Bifidobacterium lactis* (Cytogenex) significantly improved the capability of these individuals to respond to stressful situations, which has been demonstrated as increased levels of urinary free cortisol (2.4 times), a stress hormone; and DHEA-S (1.4 times), an indicator of psychological illness and stress [115]. As ME/CFS patients usually demonstrate low levels of these hormones [116], the reported increase in normal values may improve individual ability to cope with stressful events, further confirmed as significantly improved mood scores and reduced perception of fatigue on the Beck depression inventory test (BDI) (BDI-I and BDI-II) [84]. Significant reduction in anxiety among ME/CFS patients, reported after probiotic supplementation with *L. paracasei* spp. paracasei F19, *L. acidophilus* NCFB 1748 and *B. lactis* Bb12 strains, also improved their neurocognitive functioning [90].

Available studies from long COVID cohorts investigating the effects of probiotic supplementation on mental health indices are limited, as the majority focus on the management of respiratory or GI symptoms (NCT04420676; NCT04813718) after acute SARS-CoV-2 infection and/or improving immunity (NCT04922918; NCT04734886) [117]. Also, evidence of microbial-based preparations on miscellaneous symptoms still needs to be provided. To date, clinical trials dedicated to neuropsychiatric sequelae investigating the probiotic formulation (VSL#3^®^) containing eight freeze-dried strains, e.g., *Streptococcus thermophilus* BT01; *B. breve* BB02, *B. animalis* subsp. *lactis* BL03, *B. animalis* subsp. *lactis* BI04, *L. acidophilus* BA05, *L. plantarum* BP06, *L. paracasei* BP07 and *L. helveticus* BD08, after 28 days (4 weeks), have failed to improve performance and somatization symptoms and psychiatric (e.g., anxiety, depression) outcomes; nevertheless, the significant improvement in fatigue scoring may indicate that certain strains may still benefit patients despite limited effectiveness in modulating the gut–brain axis [92]. Furthermore, other studies, focused on the dietary supplementation aiming to target mental health outcomes in long COVID patients, involved preparations combining probiotic strains with other bioactives with anti-oxidant and anti-inflammatory properties, such as prebiotic fibers of fructooligosaccharides (FOS), which, added to a probiotic mixture consisting of *L. casei*, *L. acidophilus*, *L. bulgaricus*, *L. rhamnosus*, *Bacillus breve*, *B. longum* and *Streptococcus thermophilus*, have been shown to significantly decrease depressive symptoms in moderately depressed patients compared with the placebo [113]. Consequently, evidence obtained from long COVID patients, although limited by the low number of studies, has indicated that interventions with functional microbial preparations, like “OMNi-BiOTiC^®^ STRESS Repair 9”, combining nine probiotic strains, e.g., *Lactobacillus casei* W56, *Lactobacillus acidophilus* W22, *Lactobacillus paracasei* W20, *Bifidobacterium lactis* W51, *Lactobacillus salivarius* W24, *Lactococcus lactis* W19, *Bifidobacterium lactis* W52, *Lactobacillus plantarum* W62 and *Bifidobacterium bifidum* W23, with prebiotic FOS and enzymes (e.g., amylases), minerals (magnesium sulfate, manganese sulfate) and vitamin B2, vitamin B6 and vitamin B12 improved symptoms of depression and perceived emotional well-being among 44.3% (n = 27) self-assessed participants over the course of 6 months [97]. Interestingly, benefits of using synbiotics were also reported in a small recent study conducted on women with fibromyalgia co-diagnosed with ME/CFS, who, after 4 weeks of symbiotic intake (Synbiotic, Gasteel Plus^®^ (Heel Spain, S.A.U.), containing probiotic strains of *Bifidobacterium lactis* CBP-001010, *Lactobacillus rhamnosus* CNCM I-4036 and *Bifidobacterium longum* ES1 combined with prebiotics, FOS (200 mg) and micronutrients, including zinc (1.5 mg), selenium (8.25 g) and vitamin D (0.75 g), had significantly reduced anxiety when compared with the baseline score, followed by improvements in pain perception and sleep quality [96].

Given the reported associations between the alerted gut microbiome composition and the reported psychiatric complications in ME/CFS and long COVID [66], the use of psychobiotics through modulating the gut–brain axis may serve as a potential adjunct treatment to improve mental health outcomes in patients with post-viral syndromes, while also reducing stress, regulating mood and improving neurocognitive functioning [118]. Although these benefits seem to be attributed to the immunomodulatory properties of the microbial metabolites (e.g., SCFAs) [119], the limited number of studies investigating the effects of psychobiotics and their products on neurotransmitters and neurotrophic factors implicated in the central nervous system function [118] indicate a need for exploration before the use of probiotics is incorporated in clinical practice.

### 3.3. Effects of Probiotics on Gastrointestinal Symptoms in ME/CFS and Long COVID

The presence of certain gut dysbiosis reported in both ME/CFS and long COVID patients suggest that the use of microbial-based preparations (due to positive effects on gut microbiota composition) may be an efficient strategy in the management of GI symptoms in these individuals. Despite the still limited number of studies, the evidence from ME/CFS cohorts with GI complaints suggest that certain probiotic strains, especially *Lactobacilli* spp., *Bifidobacteria* spp. and/or their combination, can positively impact the gut microbiota function while promoting intestinal homeostasis, and strengthen the mucosal barrier and normalize the cytokine profile within the gut [3,45,90]. Noteworthy, an anti-inflammatory effect of probiotics by leading to the reduced production of pro-inflammatory mediators, such as CRP, TNF-α and IL-6, may have therapeutic potential for ME/CFS with comorbid IBS [45], leading to reduced IBS-like symptoms in these patients, while improving the quality of life and psychological symptoms [120]. For example, supplementation with a symbiotic formulation (Synbiotic, Gasteel Plus^®^ (Heel Spain, S.A.U.), containing probiotic strains of *Bifidobacterium lactis* CBP-001010, *Lactobacillus rhamnosus* CNCM I-4036 and *Bifidobacterium longum* ES1, in addition to prebiotics, fructooligosaccharides (200 mg), zinc (1.5 mg), selenium (8.25 g) and vitamin D (0.75 g), resulted in the better scoring of GI health indices among females co-diagnosed with fibromyalgia and ME/CFS compared with the scores reported at the baseline; nevertheless, these improvements remained not significant [96].

Similarly, for ME/CFS, the use of probiotics aiming to restore gut homeostasis has been a subject of interest, as 22% of individuals with COVID-19 diagnosis may experience certain GI distress, of which loss of appetite, dyspepsia and IBS-like and abdominal pain seem to be the most common [121]. In addition, the observed associations between changes in the composition of gut microbiome, disease severity and inflammation among long COVID patients [122,123] suggests that interventions using microbial preparations may help to reduce GI complaints among long COVID patients and improve their recovery from the disease.

To date, most studies conducted on COVID-19 survivors have shown that probiotic supplementation can improve overall symptoms, inflammatory response and time of hospitalization, with some also indicating beneficial effects on intestinal microbiota, characterized by reduced GI symptoms, e.g., reduced duration of diarrhea, abdominal pain, nausea and vomiting [69]. For example, the supplementation of a multi-strain preparation, including *Streptococcus thermophilus* DSM 32345, *L. acidophilus* DSM 32241, *L. helveticus* DSM 32242, *Lacticaseibacillus paracasei* DSM 32243, *L. plantarum* DSM 32244, *LeviL. brevis* DSM 27961, *B. lactis* DSM 32246 and *B. lactis* DSM 32247, for 7 days resulted in the reduction in diarrhea and severity of other symptoms accompanying infection (e.g., fever, asthenia, headache, myalgia and dyspnea), followed by an eight-fold reduced risk of respiratory failure among hospitalized patients with COVID-19 [94]. Although the use of probiotics might be beneficial for acute SARS-CoV-2 infection, studies assessing their efficacy in long COVID are limited. Nevertheless, an intervention with synbiotics consisting of lactobacillus strains (e.g., *Lactobacillus plantarum*, *Lactobacillus rhamnosus*, *Lactobacillus bulgaricus*, *Lactococcus lactis* and *Lactobacillus paracasei*) and prebiotic inulin after 30 days led to significant improvements in gut health and reduced GI symptoms (e.g., mild increased bloating and diarrhea), especially in older patients with a more severe course of the disease. In addition, these patients also reported significantly improved symptoms of fatigue, indigestion and cough [46]. Similar benefits were shown after the use of a multi-strain probiotic preparation “OMNi-BiOTiC^®^ STRESS Repair 9”, providing nine bacterial strains, including *Lactobacillus casei* W56, *Lactobacillus acidophilus* W22, *Lactobacillus paracasei* W20, *Bifidobacterium lactis* W51, *Lactobacillus salivarius* W24, *Lactococcus lactis* W19, *Bifidobacterium lactis* W52, *Lactobacillus plantarum* W62 and *Bifidobacterium bifidum* W23, along with prebiotic fructooligosaccharides (FOS), enzymes (amylases), minerals (magnesium sulfate, manganese sulfate) and vitamins (vitamin B2, vitamin B6, vitamin B12), which, after 3 months, led to subjective improvement in GI health, including enhanced digestion, reduced diarrhea, constipation, flatulence and abdominal pain [97]. Finally, a phase 3 clinical trial evaluating the efficacy of a probiotic formula (VSL#3^®^) containing eight strains of lactic acid bacteria and Bifidobacteria (e.g., *Streptococcus thermophilus* BT01, *B. breve* BB02, *B. animalis* subsp. *lactis* BL03, *B. animalis* subsp. *lactis* BI04, *L. acidophilus* BA05, *L. plantarum* BP06, *L. paracasei* BP07 and *L. helveticus* BD08) was shown to be effective in ameliorating GI symptoms determined by the structured assessment of gastrointestinal symptoms scale (SAGIS) compared with the placebo; however, it had limited effects on acid regurgitation, nausea, vomiting, constipation, epigastric pain and IBS-like symptoms [92].

In summary, research to date indicate that probiotics, especially lactobacillus, may be beneficial in reducing GI symptoms in patients with ME/CFS and long COVID [2,124]. The immunoregulatory properties of probiotics may help to restore the perturbed composition of gut microbiota and dysbiosis, which could reduce symptom severity (linked to the reduced abundance of butyrate producing bacteria [41]). Consequently, the use of live strains of F. prausnitzii and B. pullicaecorum, as potent butyrate producers by increasing SCFA (e.g., butyrate) levels due to its potent immunoregulatory and anti-inflammatory properties) may have to restore gut homeostasis after excessive inflammation and also, by strengthening the intestinal barrier, may prevent a leaky gut syndrome linked to IBS-like symptoms [125] resultant from increased inflammatory response/immune activation due to the leakage of pro-inflammatory endotoxins and pathogens into circulation [126]. In addition, psychobiotic and anti-inflammatory properties of certain strains may help to reduce GI symptoms resulting from psychological distress resultant from the stress-induced gut dysbiosis and activation of the HPA axis [127]. Although these effects still need to be confirmed in patients affected by post-viral syndromes, the supplementation of *L. heleveticus* Rosell-52 and *B. longum* Rosell-175 can reduce stress-induced GI discomfort among healthy populations [107]. Furthermore, probiotics have been proposed as a promising intervention for improving the health of patients with post-infectious fatigue, many of whom report GI symptoms such as abdominal pain, constipation/diarrhea or bloating. However, to confirm these effects, further studies are needed because, to date, the low number of these studies along with the overall poor quality of data do not allow to fully understand their role and effectiveness in managing GI symptoms in ME/CFS patients [2].

## 4. Future Directions

The use of microbial preparations in the management of post-viral syndromes, such as ME/CFS and long COVID, is an area of ongoing research owing to several studies indicating the role of gut dysbiosis, shared in both neuropsychiatric conditions and GI symptoms.

Reported association between low levels of butyrate and reduced abundance of SCFAs producing bacteria at the expense of gut pathogens with persisting GI and neurological symptoms in these individuals highlighted the need for novel strategies for microbiome-tailored disease prevention and treatment. The increased interest and research activity initiated during the COVID-19 pandemic, based on the certain similarity observed in the pathophysiology of both post-viral syndromes, particularly in the gut microbiome, may accelerate the progress in developing evidence-based therapeutic options for ME/CFS [13], such as the use of microbial preparations.

To date, despite limited evidence, clinical trials conducted in the ME/CFS and long COVID cohorts have demonstrated that certain microbial preparations, by restoring intestinal homeostasis, regulating the gut–brain axis and decreasing inflammation, may play a role in the management of ME/CFS and long COVID symptoms and possibly increase the quality of life of patients. Whereas certain lactobacilli and Bifidobacteria strains have been found to be effective in reducing inflammation and fatigue, their effect on the neurological and psychiatric symptoms remains inconsistent, which might be attributable to varying methodologies of the conduced trials and the low number of studies. Nevertheless, the emerging strategy of combining multi-strain probiotics with other functional ingredients previously implicated with anti-inflammatory and anti-oxidative properties, such as prebiotic fibers, vitamins and minerals and other bioactive blends, is a promising approach that may offer significant advantages for not only reducing fatigue but also increasing the well-being of patients with post-viral syndromes.

Among these strategies, formulations combining potent anti-oxidants and enzymes, such as peptidase, bromelain, amylase, lysozyme, peptidase, catalase, papain, glucoamylase and lactoferrin and vitamins and minerals implicated in energy metabolism, anti-oxidant defenses and immunity, including zinc, selenium, vitamin D, vitamin B2 (riboflavin 5′-sodium phosphate), vitamin B6 (pyridoxine hydrochloride), manganese sulfate and vitamin B12 (cyanocobalamin), may help to reduce the severity of fatigue and depression, while increasing energy levels, also improve mood, sleep and overall quality of life. Furthermore, combining probiotics with phytochemical-rich whole plant foods, such as citrus sinensis fruit, chamomile (*Matricaria recutita* L. flower), curcuma longa and pomegranate (*Punica granatum* L.) as sources of naturally derived prebiotic fibers, antioxidants and micronutrients, may provide synergistic benefits of promoting gut health and microbial activities, related to the production of microbial metabolites, which, acting systemically, can lead to improvement in overall well-being and enhance recovery. It would be interesting to assess the effectiveness of other microbial-based preparations that include postbiotics, such as microbial cells, their fractions and bioactive metabolites, such as organic acids, bacteriocins and enzymes. While using a live probiotic strain has been tested and acknowledged for its health benefits, especially in the context of immunity, gut and brain health, postbiotics, as nonviable or inactive probiotics and their metabolic by-products, may offer the significant advantage of providing all the benefits, thereby offering a potentially safer and more suitable approach for individuals with chronic inflammatory conditions and/or compromised immunity.

Although further research is needed to fully understand how postbiotics can be beneficial for ME/CFS and long COVID patients, the current evidence suggests that they could be a promising area of study for improving the health of these individuals. In particular, the use of microbial metabolites known as short-chain fatty acids (SCFAs), including acetate, propionate and butyrate, due to their immuno-modulatory properties, has been suggested to be effective in reducing the severity of COVID-19 symptoms, which might also be extended to ME/CFS, in particular for modulating the gut–brain axis and strengthening intestinal barriers [126]. For example, the supplementation of butyrate, in comparison with other SCFAs, acetate and propionate, may be the most advantageous due to its proven immunomodulatory effects, e.g., increasing secretion of mucins and defensins and promoting antiviral defenses, by inducing expression of interferon-gamma and granzyme B, and reducing systemic inflammation [126], thus possibly reducing psychiatric and fatigue symptoms. Despite these promising strategies, there is an emerging need for further studies, which would include controlled clinical trials to determine optimal formulation, doses and treatment durations, which would allow to achieve the most prominent improvements in health determinants of patients with post-viral syndromes while minimizing potential adverse effects.

## 5. Conclusions

The utilization of microbial-based preparations containing immunomodulatory probiotic strains presents a promising therapeutic approach for addressing the complex pathogenesis of post-viral syndromes, including ME/CFS and long COVID, which are also associated with comorbidities. The evidence obtained from the recent pandemic highlighted the potential of probiotics in ameliorating inflammation and GI disturbances in long COVID; however, the efficacy of these interventions in ME/CFS remains uncertain due to the limited quality of available evidence, with some studies suggesting improvements in anxiety and IBS symptoms.

While considering the pivotal role of immune dysregulation in neuroinflammation of ME/CFS patients, probiotic interventions, by improving the intestinal barriers and favorably modulating the bidirectional communication via the gut–brain axis, may help in mitigating neuropsychiatric symptoms reported in ME/CFS and long COVID. In addition, the implication of gut dysbiosis in the etiology of both conditions, including probiotic-based supplementation aiming to restore gut homeostasis could reduce the severity of symptoms associated with comorbid conditions, such as anxiety/depression, IBD, IBS and insomnia in ME/CFS. However, while these implications are promising, further research is needed to fully understand the mechanisms through which microbial-derived preparations may improve the management of ME/CFS and long COVID. Additionally, exploring the potential synergistic effects of combining probiotics/synbiotics with other bioactives with immunomodulatory and anti-oxidative properties, such as CoQ10 and selenium or CoQ10 and lipoid acid, warrants further investigation.

Taken together, a comprehensive approach utilizing a bidirectional communication via the gut–immune–brain axis indicates that dietary supplementation with microbial-based formulations combined with functional bioactives of natural origin may optimize the management of both neuropsychiatric and GI symptoms resultant from chronic inflammation and immuno-metabolic alterations, and should be considered as a part of holistic care offered to patients with ME/CFS and long COVID.

## 6. Take-Home Messages

The changes in gut microbiome reported in ME/CFS and long COVID may serve as sensors and modulators of the gut–brain axis, leading to cognitive and immune-metabolic alterations.The ability of the gut microbiome to modulate neurotransmitters and responses on the HPA axis may explain the reported associations between gut microbiota and symptom severity in ME/CFS and long COVID.Various lifestyle interventions, including diet and supplementation with microbial-based preparations (probiotics, postbiotics and synbiotics) can modulate gut microbiome activity and influence disease status.Prebiotic interventions are the most likely to result in changes to gut microbiome composition, followed by dietary interventions and then probiotic interventions in ME/CFS and long COVID.Further research is needed to validate emerging microbial metabolites and their role in ME/CFS and long COVID management.

## Figures and Tables

**Figure 1 nutrients-16-01545-f001:**
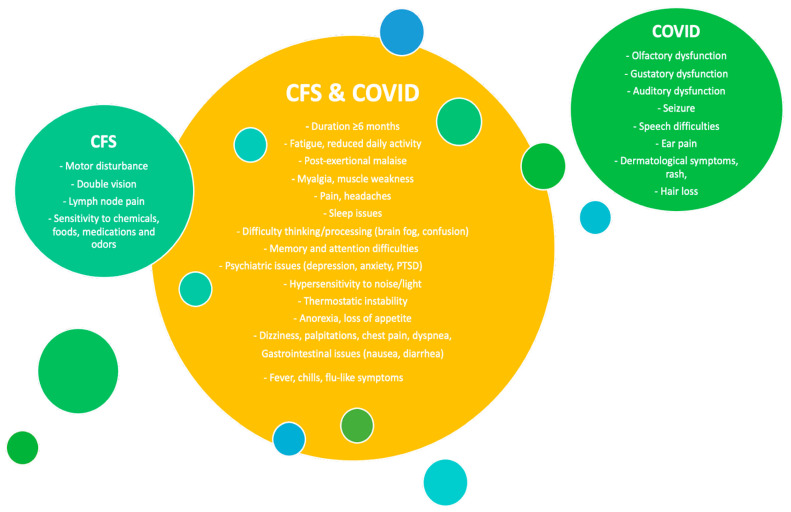
Comparison of symptoms reported in ME/CFS and long COVID. Based on [11].

**Table 1 nutrients-16-01545-t001:** Summary of studies investigating use of microbial-based preparations for management of symptoms in ME/CFS and long COVID.

Participants (n)	Intervention (Dose)		Duration	Outcomes	References
ME/CFS patients (n = 83)	Single probiotic: *Lactobacillus casei* Shirota		8 weeks	Reduced anxiety scores, changed fecal composition	[89]
ME/CFS patients (n = 83)	Single probiotic: *Bifidobacterium infantis* 35624		8 weeks	Reduced inflammatory biomarkers (CRP, IL-6)	[89]
ME/CFS patients (n = 9)	Probiotic protocol: Enterelle (given in week 1, 4–8): *Saccharomyces cerevisiae* sub. Boilardii MTCC-5375, *Saccharomyces cerevisiae* sub. Boilardii SP92, *Enterococcus faecium* UBEF41, *Lactobacillus acidophilus* LA14 Bifiselle (given in week 2): *Bifidobacterium lactis* BL04, *Bifidobacterium breve* BB03, *Bifidobacterium bifidum* BB06, *Bifidobacterium longum* BL05 Citogenex (given in week 4–8): *Saccharomyces* spp. extract titrated in 1–3 beta glucans, Vitamin C, *Lactobacillus casei* LC11, *Bifidobacterium lactis* BL04, *Lactobacillus acidophilus* LA14 Ramnoselle (given in week 3, 4–8): *Lactobacillus acidophilus* LA14, *Lactobacillus rhamnosus* LR32, *Lactobacillus rhamnosus* HN001	2 caps/a day Each strain > 1 × 10^9^ CFU	8 weeks	Reduction of fatigue on Chadler’s scale, improvement in physical and mental conditions, overall increase in quality of life on SF-36 score, improvement in mood on PCS score, reduced depression symptoms on BDI-I and II scores, reduced inflammation (CRP), improvement in immune defense (increase in IgM, reduction in CD4/CD8 ratio), inconsistent effects on oxidative status.	[84]
ME/CFS (n = 15)	Multistrain probiotic: *Lactobacillus* F19 *Lactobacillus acidophilus* NCFB 1748 *Bifidobacterium lactis* Bb12	4 × 10^8^ CFU/mL	4 weeks	Improvement in neurocognitive functions measured as the VAS mean, inconsistent effects on fatigue, mental and physical health, no major changes in GI microbiota.	[90]
Hospitalized COVID-19 patients (n = 24)	Multistrain probiotic (SLAB51; Sivomixx800^®^): *Streptococcus thermophilus* DSM 32245^®^, *Bifidobacterium lactis* DSM 32246^®^, *Bifidobacterium lactis* DSM 32247^®^, *Lactobacillus acidophilus* DSM 32241^®^, *Lactobacillus helveticus* DSM 32242^®^, *Lactobacillus paracasei* DSM 32243^®^, *Lactobacillus plantarum* DSM 32244^®^, *Lactobacillus brevis* DSM 27961^®^	3 equal doses of a total of 2.400 × 10^11^ CFU billion bacteria per day	23 days	Reduced fatigue on FAS score, minor changes in metabolomic profile, increased levels of arginine, asparagine, lactate, and decreased levels of 3-hydroxy-isobutyrate following probiotic intake.	[91]
Long COVID patients (n = 19)	Multistrain probiotic (VSL#3^®^): *Streptococcus thermophilus* BT01, *Bifidobacterium breve* BB02, *Bifidobacterium animalis subsp. lactis* BL03, *Bifidobacterium animalis subsp. lactis* BI04, *Lactobacillus acidophilus* BA05, *Lactobacillus plantarum* BP06, *Lactobacillus paracasei* BP07, *Lactobacillus helveticus* BD08	2 sashets a day, a total of 4.5 × 10^11^ CFU per sachet	4 weeks	Reduced fatigue (Chalder fatigue scale), significant improvement in physical functioning on SF-36 scale, reduction in GI symptoms on SAGIS score, no significant difference in specific symptoms of acid regurgitation, nausea and vomiting, constipation, epigastric pain and IBS symptoms vs. placebo, no effects on psychiatric symptoms of anxiety, depression, performance and somatization symptoms on SCL-12 scale.	[92]
Fatigue subjects (n = 9)	Single probiotic: *Lactobacillus paracasei* HII01	4 × 10^10^ CFU/g	12 weeks	Decreased salivary levels of cortisol, no effects on DHEA-S concentrations, decreased ratio of cortisol/DHEA-S	[93]
Hospitalized COVID-19 patients (n = 28)	Multistrain probiotic (Sivomixx^®^): Streptococcus thermophilus DSM 32345, *Lactobacillus acidophilus* DSM 32241, *Lactobacillus helveticus* DSM 32242, *Lacticaseibacillus paracasei* DSM 32243, *Lactobacillus plantarum* DSM 32244, *Lactobacillus brevis* DSM 27961, *Bifidobacterium lactis* DSM 32246, *Bifidobacterium lactis* DSM 32247	2.5 × 10^10^ CFU/a day	1 week	Amelioration of diarrhea within 3–7 days, reduction in other symptoms, including fever, asthenia, headache, myalgia and dyspnea within 2 days, improvement in respiratory function, 8-fold reduced risk of respiratory failure.	[94]
Multi-ingredient microbial preparations					
Long COVID patients (n = 100)	ImmunoSEB formulation: Probiotics-ProbioSEB CSC3: *Bacillus coagulans* LBSC (DSM 17654), *Bacillus subtilis* PLSSC (ATCC SD 7280), *Bacillus clausii* 088AE (MCC 0538) Bioactives: Peptizyme SP: enteric coated serratiopeptidase, bromelain, amylase, lysozyme, peptidase, catalase, papain, glucoamylase, lactoferrin	4 capsules a day: 1 capasule: ImmunoSEB: 500 g/capsule + ProbioSEB CSC3: 5 × 10^10^ CFU/capsule	2 weeks	Significant reduction in fatigue determined by the CFQ-11 score (87% of patients were fatigue free at the end of the intervention), significant reduction in all individual measures of physical fatigue (tiredness, need to rest, drowsiness, ability to do things, energy level, muscle strength and feeling of weakness), significant reduction in mental fatigue (concentration, focus and memory), no adverse events reported, including nausea, vomiting or diarrhea, at any stage of the study.	[86]
Long COVID patients (n = 126)	Phytochemical-rich concentrated food capsule: *Probiotic:* *Lactobacillus plantarum*, *Lactobacillus rhamnosus*, *Lactobacillus bulgaricus*, *Lactococcus lactis* *Lactobacillus paracasei* *Prebiotic:* Inulin fibre *Bioactives:* Citrus Sinensis fruit, Chamomile (*Matricaria recutita* L. flower), Curcuma Longa, Pomegranate (*Punica granatum* L.), resveratrol (*Polygonum cuspidatum* root)	1 capsule given twice a day Probiotic: 10 × 10^10^ CFU/total in capsule Prebiotic: 200 mg/capsule Phytochemical-rich whole food capsule (PC): Citrus Sinensis fruit: 400 mg, inc. 70 mg of bioflavonoids, Chamomile (*Matricaria recutita* L. flower): 1000 mg, Curcuma Longa: 23.8 mg of curcuminoid, Pomegranate (*Punica granatum* L.): 1 g, incl. 10 mg ellagic acid, Polygonum cuspidatum root: 100 mg of resveratrol	2 weeks	2-fold reduction in fatigue measured on the Chalder fatigue scale vs. placebo, reduction in other symptoms associated with infection, including cough, improved overall well-being measured on SWB score.	[95]
Fibromyalgia patients co-diagnosed with ME/CFS (n = 15)	Synbiotic (Gasteel Plus^®^) formulation: Probiotics: *Bifidobacterium lactis* CBP-001010, *Lactobacillus rhamnosus* CNCM I-4036 *Bifidobacterium longum* ES1 Prebiotic: Fructooligosaccharides Bioactives: Zinc, Selenium, Vitamin D	Each synbiotic bar (300 mg): Probiotic: 1 × 10^9^ CFU/total in bar Prebiotic: 200 mg/bar Bioactives: Zinc: 1.5 mg Selenium: 8.25 mg Vitamin D: 0.75 g	2 weeks	No changes in the objective perception of activity/sedentarism and sleep, significant improvement in depression, stress, anxiety and fatigue in FM patients, significant improvement in anxiety and fatigue in FM patients with ME/CFS, reduced inflammation (decrease in IL-8, increase in IL-10) among FM patients, increased ratio of cortisol/DHEA in all participants, overall no significant benefits for improved pain, sleep quality or gastrointestinal health of the participants.	[96]
Long COVID patients (n = 70)	“OMNi-BiOTiC^®^STRESS Repair 9” formulation: Probiotics: *Lactobacillus casei* W56, *Lactobacillus acidophilus* W22, *Lactobacillus paracasei* W20, *Bifidobacterium lactis* W51, *Lactobacillus salivarius* W24, *Lactococcus lactis* W19, *Bifidobacterium lactis* W52, *Lactobacillus plantarum* W62, *Bifidobacterium bifidum* W23 *Prebiotic:* Fructooligosaccharides, inulin *Bioactives:* Enzymes (amylases), Potassium chloride, Manganese sulfate, Vitamins B2, B6, and B12	Each sachet: 3 g Probiotic: 7.5 × 10^9^ CFU/total in sachet Prebiotic: N/A Bioactives: Enzymes (amylases), Potassium chloride, Manganese sulfate, Vitamins B2, B6 and B12	24 weeks	Significant reduction in fatigue on the FSS score vs. placebo, significant improvement in the severity of depression, significant improvement in quality of life, including physical functioning and general health, improvement in digestion and reduction in GI complaints, no effect on immune parameters	[97]

CFU—colony forming unit; n—number of participants.

## Data Availability

The original contributions presented in the study are included in the article, further inquiries can be directed to the corresponding author.

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
