# Peer review of "A Narrative Review on Gut Microbiome Disturbances and Microbial Preparations in Myalgic Encephalomyelitis/Chronic Fatigue Syndrome: Implications for Long COVID"

_nutrients, 2024, doi:10.3390/nu16111545_

Round 1

Reviewer 1 Report

Comments and Suggestions for Authors

I would like to thank you for the opportunity to read this interesting review about the role of gut microbiome in ME/CFS, and how these findings from the literature could be translated to the long COVID syndrome.

This review is very well-written and clear. I have only two comments/suggestions.

- When discussing the role of the microbiome and the link with ME/CFS (and in general, various neurological disorders), I think that it might be interesting to discuss the influence of gut microbiota on neurotransmitters (Chen et al., Nutrients, 2021). The authors have discussed it in the second section; I think that they did good work, but the section could improve if the authors provide a further connection with long COVID as it has been shown that chronic fatigue and cognitive impairment in this syndrome are characterized by altered neurotransmitters activity, as deficient GABA-B and glutamate activity in M1 (Manganotti et al., Clin Neurophysiol 2023; Ortelli et al., Eur J Neurol, 2022).

- Since the interesting and wide literature review, please consider final key points / take-home message section or bullet points, providing the reader with some practical applications based on your literature review

Author Response

Reviewer 1:

I would like to thank you for the opportunity to read this interesting review about the role of gut microbiome in ME/CFS, and how these findings from the literature could be translated to the long COVID syndrome. This review is very well-written and clear. I have only two comments/suggestions.

  1. When discussing the role of the microbiome and the link with ME/CFS (and in general, various neurological disorders), I think that it might be interesting to discuss the influence of gut microbiota on neurotransmitters (Chen et al., Nutrients, 2021). The authors have discussed it in the second section; I think that they did good work, but the section could improve if the authors provide a further connection with long COVID as it has been shown that chronic fatigue and cognitive impairment in this syndrome are characterized by altered neurotransmitters activity, as deficient GABA-B and glutamate activity in M1 (Manganotti et al. Clin Neurophysiol 2023; Ortelli et al. Eur J Neurol 2022).

We appreciate the reviewer’s insightful and helpful comments regarding our manuscript. Thank you for raising these concerns. We have addressed the impact of the gut microbiome on the disturbed metabolism of neurotransmitters in Long COVID (lines 251-260 on page 6).

  1. Since the interesting and wide literature review, please consider final key points/take-home message section or bullet points, providing the reader with some practical applications based on your literature review

We agree with this reviewers’ concerns. We have added 6 bullet points at the end of the manuscript (subheading 6: Take-home messages) as the reviewer suggested (lines 718-735 on page 20).

Reviewer 2 Report

Comments and Suggestions for Authors

In this review the authors assessed literature on the gut microbiome and microbial preparations in ME/CFS, with similarities with Long COVID. The topic is very important, as many people suffer from both conditions and the treatment is limited.

The manuscript is comprehensive, well written. I have couple of commnents and suggestions:

Title: ME and CFS needs to be first defined, like in the first line from the abstract - Myalgic Encephalomyelitis, also known as Chronic Fatigue Syndrome (ME/CFS)

Page 2, line 59 ’’...25 out of 29 known ME/CFS symptoms at least once...’’ needs a reference, so the reader may check these symptoms

Page 2 line 120 – it would be good if ''naturaceuticals'' term would be defined

Page 3, line 138, 139 – for better understanding, it would also be good to define probiotics, synbiotics, postbiotics, as these are the key elements of the review

The authors assessed studis of microbial preparations on fatigue and inflammation in ME/CFS and Long COVID, on psychiatric symptoms, gastrointestinal symptoms, but no data on cognition, brain fog etc.

There should be a section of effects of microbial preparations to miscellanious symptoms

Page 18, 19 Conclusions – I would suggest this part to be informative, without references

References needs to be double checked, as the numbers are doubled. Needs a systematic cheked, as in some cases the pages are missing

Comments on the Quality of English Language

Some minor editing in English would be necessary

Author Response

Reviewer 2: In this review the authors assessed literature on the gut microbiome and microbial preparations in ME/CFS, with similarities with Long COVID. The topic is very important, as many people suffer from both conditions and the treatment is limited.  The manuscript is comprehensive, well written. I have couple of comments and suggestions:

  1. Title: ME and CFS needs to be first defined, like in the first line from the abstract - Myalgic Encephalomyelitis, also known as Chronic Fatigue Syndrome (ME/CFS).

Thanks to the reviewer for raising this key point. It has been addressed now (lines 3-4 on page 1)

  1. Page 2, line 59: “... 25 out of 29 known ME/CFS symptoms at least once...’’ needs a reference, so the reader may check these symptoms.

Thank you for your suggestion. Reference has been added (Wong TL et al. Medicina 2021; 57(5):418. doi: 10.3390/medicine57050418)

  1. Page 2, line 120: It would be good if ''nutraceuticals'' term would be defined.

Thank you for raising this key point. We have removed “nutraceutical” as reference here highlights of use micronutrients and other bioactives, however does not state about the origin – now the statement is more indicative (lines 124-127 on pages 3-4)

  1. Page 3, lines 138-139 – for better understanding, it would also be good to define probiotics, symbiotic, postbiotics, as these are the key elements of the review.

These concepts have been defined in the text (lines 129-136 on pages 3-4)

  1. The authors assessed studies of microbial preparations on fatigue and inflammation in ME/CFS and Long COVID, on psychiatric symptoms, gastrointestinal symptoms, but no data on cognition, “brain fog”, etc.

Thank you for this concerns from reviewer. Based on the search results provided, the use of microbial preparations, specifically probiotics, has shown some promise in reducing miscellaneous symptoms, cognition, and "brain fog" in ME/CFS. There is currently no evidence that microbial-derived preparations can mitigate the miscellaneous symptoms including cognitive impairments (aka “brain fog”) in Long COVID. We have added this minor information to text accordingly (lines 481-486 on page 15)

  1. There should be a section of effects of microbial preparations to miscellaneous symptoms.

We agree. Unfortunately, there are not enough data/evidence (limited information) to add a new section on the effects of microbial preparations on miscellaneous symptoms in ME/CFS and Long COVID.

  1. Conclusions on pages 18-19: I would suggest this part to be informative, without references.

Done.

  1. References needs to be double checked, as the numbers are doubled. Needs a systematic checked, as in some cases the pages are missing.

It has been checked accordingly.

Round 2

Reviewer 2 Report

Comments and Suggestions for Authors

Authors addressed all my questions and comments. Thank you.